# Effect of a Simulated Match on Lower Limb Neuromuscular Performance in Youth Footballers—A Two Year Longitudinal Study

**DOI:** 10.3390/ijerph17228579

**Published:** 2020-11-19

**Authors:** Michal Lehnert, Mark De Ste Croix, Amr Zaatar, Patrycja Lipinska, Petr Stastny

**Affiliations:** 1Faculty of Physical Culture, Palacky University Olomouc, 771 11 Olomouc, Czech Republic; amr.zaatar@upol.cz; 2School of Sport and Exercise, University of Gloucestershire, Gloucester GL50 2RH, UK; mdestecroix@glos.ac.uk; 3Institute of Physical Education, Kazimierz Wielki University in Bydgoszcz, 85-064 Bydgoszcz, Poland; patlipka@gmail.com; 4Faculty of Physical Education and Sport, Charles University, 162 52 Prague, Czech Republic; stastny@ftvs.cuni.cz

**Keywords:** simulated match-play, leg stiffness, reactive strength, EMG, isokinetic

## Abstract

The aim of this study was to explore the effects of simulated soccer match play on neuromuscular performance in adolescent players longitudinally over a two-year period. Eleven players completed all measurements in both years of the study (1st year: age 16.0 ± 0.4 y; stature 178.8 ± 6.4 cm; mass 67.5 ± 7.8 kg; maturity-offset 2.24 ± 0.71 y). There was a significant reduction in hamstring strength after simulated match by the soccer-specific aerobic field test (SAFT^90^), with four out of eight parameters compromised in U16s (4.7–7.8% decrease) and six in the U17s (3.1–15.4%). In the U17s all of the concentric quadriceps strength parameters were decreased (3.7–8.6%) as well as the vastus lateralis and semitendinosus firing frequency (26.9–35.4%). In both ages leg stiffness decreased (9.2–10.2%) and reactive strength increased pre to post simulated match (U16 8.0%; U17 2.5%). A comparison of changes between age groups did not show any differences. This study demonstrates a decrease in neuromuscular performance post simulated match play in both ages but observed changes were not age dependent.

## 1. Introduction

During soccer match-play a significant amount of sudden accelerations and decelerations, jumping and landing tasks, and rapid unanticipated changes of directions are performed [1,2,3]. These high-intensity movements may be associated with increased predisposition to injury risk in youth players [4], especially when fatigue is likely present at the end of both the first and the second halves of match-play [5]. The most common injuries in both adult and youth soccer are reported in lower limbs, specifically the hamstrings, knee joint, and ankle [6,7,8,9,10]. A recent systematic review on youth players concluded that the injury incidence in young soccer players has increased, with values exceeding those of professional soccer players, mainly in muscle injuries [6,9]. It has been well documented that local fatigue is one of the main etiological factors which contributes to lower limb noncontact injuries in soccer [11]. Local fatigue, both as a result of acute load or its long-term application, increases the risk of injury, alters muscular activation and coactivation, lower limb kinematics, reactive strength, muscle stiffness and other factors associated with injuries and soccer performance [12,13,14,15,16].

The muscle antagonist force production capacity is considered an essential factor for joint stability during intensive movements [17,18,19,20], where ipsilateral knee flexors (hamstrings, H) and extensors (quadriceps, Q) strength imbalance and fatigue are included in ACL injuries [18,19,21,22,23] and hamstring strain injuries [18,24,25,26]. Although there is a lack of evidence on the predictive value of H/Q ratios for ACL injuries, it was demonstrated that soccer players with isokinetic strength imbalance were 4.66 more likely to sustain hamstring strain injury [27]. The cause of hamstrings injuries is often related to altered neuromuscular coordination running patterns induced by fatigue [28]. Adult studies have reported that submaximal muscular fatigue increases anterior tibial translation along with hamstring latency, which influence knee joint stability by ligament and tendon interactions [29,30]. Available studies of soccer-specific fatigue on neuromuscular function predominantly point to the reduction in muscle strength after physical load, where soccer sprints and deceleration ability correlates with the H and Q concentric and eccentric strength at different speeds [31,32]. In some studies, reduction of the H strength when fatigue is present [11,33] and decrease in the H/Q have been observed [11,25,34,35,36]. However, conflicting data are available as other studies on youth soccer players have shown no significant decrease in the H/Q, but leg stiffness and RSI were compromised [37,38,39]. Activity of muscles which help to stabilize the knee joint may be assessed by means of surface electromyography (EMG). Research has identified that well-timed activation of the H muscles can protect the ACL from mechanical strain by stabilising the tibia and reducing anterior tibial translation and that the speed of this activation is vital for the subsequent joint stability [40]. It is also accepted that analysis of EMG signal by mean frequency is a feasible method of assessing changes in muscle activation due to fatigue [41,42]. Only two studies appear to have explored changes in neuromuscular function, using EMG, in young soccer players following simulated match-play [37,38] and their findings point to fatigue induced changes in activation in knee flexors and extensors of players. In one study [43], the effects of competitive match-play on the neuromuscular function was explored in adolescent soccer players and no effects of match play on muscle activation were evident both for knee flexors and extensors. 

Leg stiffness, incorporating muscle, tendon, and joint stiffness, is also based on neuromuscular feed-forward mechanism, acting as a protective mechanisms against injury to counteract the external forces placed on the tibiofemoral joint, particularly during dynamic movements to improve joint stability [19,44]. Greater levels of leg stiffness leads to a reduction in the probability of excessive load of the knee passive structures such as the ACL [19]. There are only a few studies examining the influence of simulated match-play on leg stiffness in adolescent soccer players showing both a decrease [38,45] and no change [46]. Reactive strength, assessed by means of the reactive strength index (RSI), has been suggested as a reliable measure of stretch-shortening cycle (SSC) capability in youth athletes [47]. RSI has been developed as a tool to monitor stress on the muscle-tendon complex during plyometric exercise and is described as the ability of transition from eccentric to concentric muscle actions [48,49]. Low values of RSI are considered a potential risk factor in ACL injury [39,50]. Likewise in the case of stiffness, there is minimal research on changes of RSI after soccer specific fatigue in youth male soccer players; however, most studies have indicated negative fatigue related effects on reactive strength [38,45,46].

It is well recognized that the risk of injury affects both male and female athletes of all ages, however, this risk increases during growth and maturation [6,51,52] and is linked to periods of rapid growth (Peak height velocity) [53] typically in 13–15 year-olds [54]. Other predisposed ages are 16–18 and 9–12 year-olds [54,55]. In footballers of this age group, 70% of all leg injuries are in the lower extremities [54]. In a recent study De Ste Croix et al. [56] reported a reduction in peak torque (PT) after a simulated match by the soccer-specific aerobic field test (SAFT^90^) in all female youth players irrespective of age, but this resulted in only a small effect of fatigue on neuromuscular function determined by changes in muscle activation and electromechanical delay. However, in a similarly focused study on female youth soccer players (13–15 y), the change of the H/Q_FUNC_ after the SAFT^90^ was observed with age related differences [57]. Further, significantly longer electromechanical delay was observed in children compared with adults [58] and developmental changes were observed in a study on female youth players [56]. This suggests that children and adolescents are an at risk group in terms of injury but also highlights the lack of data on male youth players [59].

There is lack of research regarding how acute post exercise fatigue influences physiological mechanisms in youth populations, including adolescent soccer players, and no studies appear to have examined the acute effects of simulated soccer match play during growth using longitudinal data. As changes in neuromuscular function, which has been associated with injury risk, are modifiable, it is important to understand the age-related effects of fatigue on neuromuscular mechanisms. This knowledge enables coaches to better prepare players by applying intervention training programs to improve fatigue resistance and at the same time positively influence performance capacity of players. Therefore, the aim of this study was to explore the effects of simulated soccer match play on the neuromuscular performance in adolescent players and examine if changes differ in male adolescent soccer players within a one year time period. 

## 2. Materials and Methods 

### 2.1. Participants

A group of eleven elite youth footballers, playing the highest Czech league in their category, were measured in the 1st and 2nd year of the study (19 recruited, 8 dropped, 11 remained). The 11 players were under 16 years of age (U16) during the 1st year and under 17 years (U17) in the second year (Table 1). The training age of the players was 8–9 y. The players played one competitive match and trained five times per week (7–8 h) during both seasons. Training consisted of typical age-related training for youth athletes: i.e., physical fitness training (mainly strength and power, speed and agility with and without ball, repeated sprint ability with and without ball), skill-oriented training (technical-tactical training, game-like training), and recovery training.

Only healthy players who had not sustained a serious musculoskeletal lower-extremity injury (in this study defined as an injury which caused an interruption of the regular training process for more than two weeks) in the previous six months were included in the study. Goalkeepers were not included in the study. The kicking leg was determined by kicking preference and subsequently the other limb was determined as the stance leg. Players and their parents were fully informed about the aim of the study and the testing procedures. The study protocol and written informed consent was approved by the Palacky University institution’s ethical committee is in accordance with Declaration of Helsinki 2013. Written informed consent to the testing procedures and the use of the data for further research was obtained from the players’ parents and written informed assent from children. The day before testing, the players were not exposed to any high intensity exercises. All players were on time maturing after the peak height velocity (PHV). Biological maturity was calculated as offset from PHV using the Mirwald et al. [60] equation. 

### 2.2. Procedures

The players were longitudinally tested using a repeated-measures design towards the end of the competitive season (Figure 1A). 

A habituation session took place a week before testing, where players were familiarised to the soccer specific protocol (SAFT^90^, Figure 1B), isokinetic dynamometry and anthropometric measures, including leg length, tibia length, and sitting height for the purpose of PHV offset calculation, were taken. During the first and second session, the players completed a warm-up consisting of 5 min cycling at 1.5 W∙kg^−1^, 6 min of dynamic stretching of the lower limbs, and fifteen squats. The main testing consisted of measuring the leg stiffness, RSI, and isokinetic dynamometry with integrated EMG. These tests were consistently undertaken in the above-mentioned order pre and post the SAFT^90^. Both sessions were conducted at the university research laboratory by experienced researchers and research assistants. 

### 2.3. Isokinetic Dynamometry

The isokinetic strength of the kicking leg and stance leg was measured using an dynamometer (IsoMed 2000; D. & R. Ferstl GmbH, Hemau, Germany) with high reproducibility in sporting populations [61] for concentric and eccentric actions and with gravitational correction for the mass of the measured lower limb. Participants were placed in a seated position with a hip flexion angle of 100° with strapping placed over the pelvis, shoulders, and thigh of the tested leg. The lever arm of the dynamometer was aligned with the lateral epicondyle of the knee and with the distal part fixed to the shin ≈ 2 cm above the medial malleolus. 

The testing protocol included concentric and eccentric single actions of knee flexors and in concentric single actions of knee extensors at 60°∙s^−1^ and 180°∙s^−1^ in the 10–90° range of knee flexion (0° = full voluntary extension). The protocol included two sets of 3 maximum repetitions at each velocity with 1 min rest. Visual feedback was provided by participants observing the strength curve, accompanied by strong verbal encouragement. For the assessment of changes and differences in isokinetic strength, peak torque normalized for body weight (relative PT; Nm∙kg^−1^) was used. To calculate the functional hamstring: quadriceps ratio (hamstring eccentric to the quadriceps concentric; H/Q_FUNC_) and conventional hamstring: quadriceps ratio (hamstring concentric to the quadriceps concentric; H/Q_CONV_) absolute peak torque (PT; Nm) was used. The reliability of H/Q ratios has been reported as being medium to high (H/Q_CONV_: ICC = 0.73; H/Q_FUNC_: ICC = 0.62) [62]. 

### 2.4. EMG 

Surface EMG data were obtained using an 8-channel polyelectromyography (Noraxon-Myosystem 1400A, Scottsdale, AZ, USA) during the isokinetic task described above. The within-session reliability of EMG during isokinetic knee actions has been reported as high (ICC = 0.86) with the between-session reliability lower (ICC = 0.65) [63]. We used Kendall-ARBO silver chlorid electrodes with a solid hydrogel with a diameter of 24 mm. The signal was captured eight manifolds with a 1000 Hz frequency. The resistance of the poly-EMG device was >10 MΩ (24 mm). Prior to application of the electrodes, the skin in the area was cleaned with water and dried. Surface electrodes were placed on the muscle belly in parallel with the process of muscle fibres with 1 cm distance between electrodes, with a ground electrode located on the tibial tuberosity. For the second measurement if the electrode was dislodged due to sweating during the fatigue protocol then a new electrode was used to reduce impedance and improve the electrode contact with the skin [64]. Electrodes were placed on the kicking leg on the biceps femoris (BF), semitendinosus (ST), vastus lateralis (VL), vastus medialis (VM), lateral gastrocnemius (LG), and medial gastrocnemius (MG). EMG data were analysed in the MyoResearch XP Master Version 1.03.05 Noraxon software (Noraxon-Myoresearch, Scottsdale, AZ, USA), where the raw EMG signal was full wave rectified and smoothed. Data for each movement velocity and muscle were split into the resting phase and muscle action phase and the mean frequency value (Hz) was determined for each phase. For normalization of the signal, retained ratio between the value of the mean frequency in the resting phase and the value of the phase of muscle activity was used.

### 2.5. Measurements of Reactive Strength Index

The RSI was measured using a 30 cm drop jump test with hands positioned on the hips using an Opto-jump Next system (Microgate, Bolzano, Italy) with 0.001 s accuracy. Participants were instructed to perform a maximal jump utilizing as short a take-off phase as possible [65]. The greatest value from three trials was used for further analysis, and RSI was calculated as the ratio between jump height and contact time [47,52]. This method has been shown to be valid and reliable in youth athletes [47].

### 2.6. Leg Stiffness

Leg stiffness was calculated from contact time data obtained during a 20 submaximal bilateral hopping test with a coefficient of variation in youth soccer players of 8.2% [66]. Participants performed three sets of consecutive hops on force platform PS-2142 (Pasco, Roseville, CA, USA) at a hopping frequency of 2.5 Hz [47] determined by a quartz Wittner metronome (WITTNER, Isny, Germany). 

Participants were instructed to jump and land on the same spot, to keep hands on the hips, land with legs fully extended, and to look forward at a fixed posture. The first 4 hops were discounted and the 10 hops closest to the hopping frequency were used in subsequent analysis. Absolute leg stiffness (kN·m^−1^) was calculated by using the Dalleau et al. [65] equation and divided by body mass to determine relative leg stiffness [47,66]. This method has been shown to be valid and reliable in youth athletes [47,66].

### 2.7. Fatigue Protocol

The SAFT^90^ which incorporates frequent acceleration and deceleration typical of match-play was used [35]. The SAFT^90^ was created according to data from English Championship matches (Prozone^®^) and was validated to replicate the fatigue response of soccer match play [35]. The SAFT^90^ duration was 2 × 40 min with 15 min rest and intensity and type of activity during the test were maintained by prerecorded 15-min verbal signals from MP3 player. 

### 2.8. Statistical Analysis

Descriptive statistic and t test were performed using Statistica, version 12 (StatSoft, Inc., Tulsa, OK, USA) and changes in pre to post match data were interpreted using a magnitude-based decision method (MBD) [67], according to current recommendation by Greenland [68]. Analyses were performed using the spreadsheet available online ([67], https://www.sportsci.org/2019/) for the analysis of post-only control trials with adjustment for a predictor [69], which also permits analysis of data consisting of repeated measurements of subjects in one group [70]. 

The data and magnitudes of the effects were log-transformed and standard deviations of mean change scores were back-transformed to percent units as 0.2, 0.6, 1.2, and 2.0 for small, moderate, large, and very large effect respectively. Uncertainty was expressed as 90% confidence limits and as probabilities that value of the effect was beneficial, harmful, or trivial. These probabilities were presented as clinical inferences [69]. Unclear effects were identified as odds ratio of benefit to harm <66. Clinically clear effects were 25–75%, possibly; 75–95%, likely; 95–99.5%, very likely; >99.5%, most likely. Wilcoxon pair-tests was used to examine whether changes in the variables differed between age groups. The effect sizes were determined by the r coefficient and evaluated as small (r = 0.1), medium (r = 0.3), and large (r = 0.5) [71].

## 3. Results

The pre- and post- SAFT^90^ values for all measured variables for groups over the two testing occasions and the results of magnitude-based decision statistics are reported in Appendix A. For technical reasons, EMG data was not obtained from two players in the second year of measurement and isokinetic data from one player in the first year of measurement.

The SAFT^90^ protocol decreased H strength in both groups U16 and U17. Four out of eight H strength parameters were decreased in U16 and six out of eight in U17 age groups. All of the concentric Q strength parameters were decreased in U17 and none in U16 (while one Q strength value was increased in the U16 group) (Figure 2 and Figure 3). 

Specifically, the SAFT^90^ induced a reduction in concentric H strength at 180°∙s^−1^ in the kicking leg, at 60°∙s^−1^ in the stance leg and in eccentric H strength at 60°∙s^−1^ in both limbs in the U16 group. The effect ranged from possibly harmful to likely harmful. Q concentric strength at 180°∙s^−1^ in the stance leg significantly increased (likely beneficial) but no change in all other Q strength parameters were observed in the U16 group. Consequently, the H/Q_CONV_ at 60°∙s^−1^ in the stance leg decreased in the U16 group post-SAFT^90^ (likely harmful effect). After the SAFT^90^, a significant decrease in the concentric H strength in stance leg at both velocities, in eccentric H strength at 60°∙s^−1^ in both limbs and at 180°∙s^−1^ in the kicking leg (the effect ranged from most likely harmful to very likely harmful) was observed in the U17 age group. Concentric Q strength decreased at both velocities and in both limbs. The effect ranged from most likely harmful to very likely harmful. After the SAFT^90^, an increase in the H/Q_CONV_ at 60°∙s^−1^ for the kicking leg was also observed in the U17 age group.

In both the U16 and U17 groups leg stiffness decreased and RSI increased (Figure 2 and Figure 3) after the SAFT^90^. For both groups, the magnitude of the effect was very likely harmful for both absolute and relative leg stiffness. Changes in RSI were very likely beneficial in the U16s and possibly beneficial in U17s. 

Increased muscle activation of the ST during concentric knee flexion at 60°∙s^−1^ and at 180°∙s^−1^ (the effect was likely beneficial in both cases) and increased activation of the BF during concentric knee flexion at 60°∙s^−1^ (the effect was very likely beneficial) (Figure 4) was observed in the U16 group after the SAFT^90^. 

Muscle activation of the plantar flexors demonstrated a decrease in LG at 60°∙s^−1^ during eccentric knee flexion actions (very likely harmful effect) and at 60°∙s^−1^ in both concentric and eccentric knee flexion action (both possibly harmful effect). In the U17 a decreased muscle activation of the ST during concentric and eccentric knee flexion at 60°∙s^−1^ (the effect was very likely harmful), and a decrease in the VL during concentric knee extension at 60°∙s^−1^ and 180°∙s^−1^ (the effect was likely harmful and very likely harmful) (Figure 4) was observed after the SAFT^90^. Moreover, muscle activation of the plantar flexors during eccentric knee flexion at 60°∙s^−1^ and 180°∙s^−1^ decreased following fatigue exercise (Appendix A). The effect was likely harmful and most likely harmful for LG and likely harmful in both the cases in MG.

A comparison of the differences in the changes of the observed variables in the pre and post-test values between both age groups confirmed significant differences only in H eccentric PT at 180°∙s^−1^ for the stance leg (*p* = 0.026; r = 0.473), Q concentric PT at 180°∙s^−1^ for the stance leg (*p* = 0.016; r = 0.512), H/Q_CONV_ at a velocity of 180°∙s^−1^ (kicking leg: *p* = 0.032; r = 0.454; SL: *p* = 0.032; r = 0.454), and muscle activation in the case of BF (*p* = 0.022; r = 0.530).

## 4. Discussion

The aim of this study was to explore the effects of simulated soccer match play on the neuromuscular performance in adolescent players longitudinally over a two-year period. The finding of this study indicates that simulated soccer match play induces a significant decrease in HPT and leg stiffness and increase in reactive strength, irrespective of participants age. However, greater changes/effects in neuromuscular function and performance were found in U17 compared to U16 age. The U17 also demonstrated significantly greater reductions in firing frequency of motor units in VL and BF, compared to the U16. These results do not indicate a positive influence of age on fatigue resistance in observed youth soccer players.

### 4.1. Changes in Torque Production in Hamstrings

Local muscular fatigue is a factor associated with injury which can explain the greater risk of injury in the second half of a soccer game and towards the end of each half [72]. In the case of muscle strength, in this study fatigue produced by the SAFT^90^ was defined as a reduction in relative PT production by muscle group. Changes in muscle strength resulted in a reduction (likely or very likely harmful, except for H eccentric PT at 180°∙s^−1^ for the stance leg) of PT production of knee flexors (hamstrings) in particular during eccentric actions in the U17 (Figure 3) and partly in U16 groups (likely harmful in H eccentric PT at 60°∙s^−1^ in both legs) (Figure 2). The decline in H strength (both eccentric and concentric) may be explained by the efforts of the H in the control of running activities and for stabilizing the knee joint during foot contact with the ground [25]. The reduction of hamstrings PT production in eccentric action has an implication in practice because H strain occurs predominantly during the latter part of the swing phase of sprinting when H work eccentrically and their tension is maximal to stabilize the knee [73]. This result could also indicate that after simulated soccer, susceptibility to ACL injury increases in players due to loss of H muscle strength and consequently deterioration of their stabilization function [74]. This capacity for muscular knee joint stabilization is progressively augmented at gradually more extended knee joint positions and increasing angular velocity.

These changes are consonant with findings in a recent study [35], where only H PT during eccentric actions were reduced significantly (16.8%) at a velocity of 120°∙s^−1^ in youth soccer players after a fatigue protocol. In addition, in a study on female youth soccer players, a reduction in PT was evident after the SAFT^90^ irrespective of age (concentric PT 17% and eccentric PT 26%, respectively [56]. 

A significant difference, with a medium effect size, in changes of PT in the H between both age groups was observed only for H eccentric PT at 180°∙s^−1^ for the stance leg (*p* = 0.026, r = 0.473), with a nonsignificant increase of PT (*p* = 0.657, r = 0.090) in the younger age group and a nonsignificant decrease (*p* = 0.286, r = 0.208) in the older age group. Although a different direction of change was observed in this case, other results suggest that chronological age does not influence the ability of the H to resist fatigue after simulated match-play. 

### 4.2. Changes in Torque Production in Quadriceps

Knee extensor (quadriceps) strength during concentric muscle actions is compromised significantly in older players (in younger players, a decrease in only one strength parameter was observed) with the effects of the reductions observed ranging from possibly harmful to very likely harmful (Figure 3). Due to the stabilization function of the Q during contact phase, this tendency, together with the significant changes in strength in both age groups could point to an impairment in the ability of these muscles to help stabilise joints after simulated soccer in youth players. The reduction of torque production in both Q and could be attributed to peripheral fatigue caused in particular by depletion of muscle glycogen stores, creatine phosphate concentration, dehydration, and decreased ability to recruit muscle fibres [25]. However, to what extent peripheral or central fatigue contributed to this reduction, could not be assessed in this study. 

A comparison of the differences in changes of PT in Q strength between age groups indicated that differences were significant only in the VL at 180°∙s^−1^ (*p* = 0.016, r = 0.512), with a significant decrease of PT (*p* = 0.032, r = 0.456) in the U17 group and nonsignificant increase (*p =* 0.248, r = 0.235) of PT in the U16 group. This indicates that the change in Q strength is not age dependent.

### 4.3. Changes in H/Q Ratios

Acute fatigue changes in H/Q ratios resulted only in a likely harmful decrease of the H/Q_CONV_ at 60°∙s^−1^ stance leg in the U16 group and likely beneficial increase in the H/Q_CONV_ at 60°∙s^−1^ in the kicking leg in the U17. However, from the point of view of injury risk, we do not consider the increase in the U17 as beneficial since it was achieved by a decrease of Q strength and not an increase in H strength. These results do not indicate impairment of muscular control postfatigue in observed players of both age groups. This suggestion is also supported by the results of H/Q_FUNC_, where the average value in both measurements was around and/or above 0.7. Findings of the latest review by Baroni et al. [75] support reference value close to 80% for isokinetic H/Q_FUNC_ ratio at slow angular velocities up to 60°∙s^−1^ as an indicator of injury risk in professional male soccer players. These results support the results of a previous study by Lehnert et al. [38] on male soccer players aged U15, where no fatigue related decrease in the H/Q_FUNC_ at angular velocities of 60°∙s^−1^, 180°∙s^−1^, and 360°∙s^−1^ were observed after the SAFT^90^. On the contrary, these results contradict the results of studies in adult soccer players which indicated decline in H/Q ratios after simulated soccer match play [25,76]. In one study [76], both H/Q_CONV_ and H/Q_FUNC_ were significantly reduced after exercise only in a group of players with balanced H/Q ratio (>0.60). However, differences in the results in previous studies on youth players and adult players could be explained by the differences in the exercise used to induce fatigue as well as the participants investigated.

As far as a comparison of differences in changes of H/Q ratios in pre and post-test values (before and after SAFT^90^) between both age groups is concerned, significant differences were found only in H/Q_CONV_ at a velocity of 180°∙s^−1^ with a medium effect size (kicking leg: *p* = 0.032, r = 0.454; stance leg: *p* = 0.032, r = 0.454), with a nonsignificant decrease observed in both age groups (U16, kicking leg: *p* = 0.308, r = 0.208; stance leg: *p* = 0.328, r = 0.199; U17, kicking leg: *p* = 0.722, r = 0.075; stance leg: *p* = 0.859; r = 0.037). These results support the findings of the changes in PTs in the current study and do not indicate an influence of age on fatigue resistance in observed players.

### 4.4. Changes in EMG

The EMG data demonstrates a significant decrease in muscle activation after the SAFT^90^ compared to the nonfatigued state, especially in the older age group (U17). There were significant reductions at a velocity of 60°∙s^−1^ in knee flexors (ST: very likely harmful, concentric and eccentric muscle action) and extensors (VL: very likely or likely harmful, concentric muscle action during knee extension) and also in plantar flexors (LG: most likely harmful, eccentric muscle action during knee flexion). This data suggests that toward the end of match play, youth players’ ability to effectively utilize neuromuscular mechanisms to control joint movement and reduce load on ligaments is reduced and injury risk is increased. In U16 players acute fatigue changes in muscle activity resulted only in a very likely harmful decrease in LG (eccentric muscle action during knee flexion) at a velocity of 60°∙s^−1^ and 180°∙s^−1^. In contrast, we observed very likely beneficial changes in BF (concentric flexion; 60°∙s^−1^) and likely beneficial changes in ST concentric flexion; 60°∙s^−1^ and 180°∙s^−1^). 

The findings in the U17 are to some extent in agreement with one previous pediatric study that demonstrated a significant increase in electromechanical delay following the SAFT90 in female youth soccer players [56]. However, in our study a significant decrease of mean frequency values was localised in different muscle groups and was observed in more muscle groups. This fatigue related effect in knee flexors (ST, concentric flexion) and also extensors (VL, concentric extension) at the U17 supports the findings of a previous study on youth footballers aged U15 [38], where the authors reported significantly compromised muscle activity in ST at 60°∙s^−1^, 120°∙s^−1^ and nonsignificant decrease at 180°∙s^−1^. In the BF there was a trend of a reduction; however, this was nonsignificant. Surprisingly, in the most recent study by De Ste Croix et al. [43], which explored the effects of competitive soccer match-play on neuromuscular performance and muscle damage in 13-16y players (further split into PHV group and post PHV group based on maturity off-set), no significant changes in muscle activation of H and Q muscles were observed. Considering all the above-mentioned studies in which elite youth soccer players were observed, the differences in changes of muscle activity could be, in our opinion, explained by differences in the applied fatigue exercise (soccer-specific fatigue protocol and competitive match-play, respectively). However, this is only speculation; the workload was not monitored in any of these studies.

An increase in injury risk of the lower limbs after the SAFT^90^ in the older age group (U17) may be also due to a significant decrease in mean frequency in the VL (at both velocities). VL and MV act as agonists (extension of knee joint) and also as antagonists (control position of patella). Based on the reduction in the VL, we could assume that the MV is weaker in players and thus VL replaces its function, which can affect the position of the patella laterally. Furthermore, we have seen a significant decrease in the mean frequency value of LG during eccentric muscle actions of the knee flexors in both age groups (at 16 y the reduction was at 60°∙s^−1^ while at 17 y at both measured velocities). Despite the fact that the function of LG for stability of the knee joint is secondary, a decrease in LG activity may have negative consequences for injury risk during landing, cutting, and running. 

The between group differences in changes of muscle activation were found for both the knee flexors and extensors. The SAFT^90^ increased the knee flexors (ST and BF) muscle activation in the younger age group and decreased knee flexors (only ST) muscle activation in the older age group. However, the change was statistically significant only in the case of BF (*p* = 0.022; r = 0.530). Furthermore, the SAFT^90^ decreased muscle activation of VL during concentric knee extension at both velocities in the U17s and resulted in no change in U16s; however, the difference in changes was not significant. The explanation of the higher (although mostly nonsignificant) decrease of muscle activation and compromised state of most of the other observed injury risk indicators after the SAFT^90^ in the older age is difficult. However, one of the reasons could be the difference in anthropometric characteristics, especially a lower body mass. It could also be related to the difference in the training and competition load in recent training cycles and higher level of fatigue. Nevertheless, the players were not exposed to a high intensity training load the day before testing in both years. Unfortunately, the current level of fatigue was not monitored in players. In this context, interesting results are presented in the already mentioned study on youth soccer players aged 13–16 y by De Ste Croix et al. [43]. The authors reported that the influence of competitive match-play on neuromuscular function evaluated by muscle activity recorded using surface polyelectromyography and RSI was found to be similar in male youth around the time of peak height velocity and those post peak height velocity. The authors suggested that these results indicate that other factors must contribute to the heightened injury risk around PHV.

### 4.5. Changes in Leg Stiffness

Both U16 and U17 groups demonstrated a very likely harmful decrease in absolute and relative leg stiffness after the SAFT^90^. Compromised stiffness when fatigue is present may indicate fatigue-induced changes in muscle-tendon complex activation and consequently reduced ability to produce muscle strength and resist against deformities (absorb energy) originated during SSC [12]. The potential consequence of such reductions could be an increase in shear force absorption directly by the knee joint, which would have negative permeations for ACL injury risk [77]. Moreover, decreased leg stiffness negatively influences jump and speed performance [78]. From a neuromuscular perspective, it is likely that fatigue will have induced a change in the activation of the musculotendon unit, leading to a reduction in preactivation prior to ground contact (feed-forward control) and an increase in cocontraction after ground contact (feedback control). Up to 97% of the variance in leg stiffness has been explained by the contribution of preactivation and stretch-reflex response of lower limb extensor muscles [59]; therefore, changes in stiffness are likely to reflect changes in these control mechanisms.

Mechanically, a reduction in leg stiffness would typically be characterized by an increased yielding action, greater ground contact times, greater centre of mass displacement, and less efficient movement when the limb comes into contact with the ground [79]. The potential consequence of such fatigue-induced reductions in ground reaction forces (and overall leg stiffness) could be an increase in shear force absorption directly by the knee joint, which would have negative permeations for ACL injury risk [77]. A reduction in leg stiffness in the measured age groups may place an individual at increased risk of lower limb injury due to a reduction in dynamic stabilization of the knee.

Only a small number of studies appear to examine changes of leg stiffness after specific fatigue protocol and/or a competitive soccer game. Our findings are in agreement with one newer study on 14-year-old players [38] where a significant reduction in both absolute and relative stiffness was reported after the SAFT^90^. In addition, in a study on 15-year-old players [45], absolute leg stiffness deteriorated nonsignificantly after soccer-specific fatigue protocol. The authors, however, point out that in half of players the stiffness increased and in the other half it was reduced. In contrast, in the study by De Ste Croix et al. (2019) on 13–16 y old soccer players, the authors observed no significant fatigue-related change in absolute and relative leg stiffness both in a PHV group and a post-PHV group. The authors also state that their findings support some of the previous literature in youth players that have also identified that acute changes in leg stiffness appear to be very individualized. Surprisingly, an increase in the relative leg stiffness was recorded after a soccer-specific exercise in 16-year-old female youth players [66].

Comparison of differences in changes of leg stiffness between both age categories showed that differences were not age dependent (*p* = 0.094 and r = 0.075). In both the groups, the decreased values after the SAFT^90^ indicate that due to the fatigue state, neural control in knee was reduced and ACL injury risk increased [80,81]. 

### 4.6. Changes in RSI

After the SAFT^90^, RSI demonstrated very likely beneficial changes in U16 and possibly beneficial changes in U17. As RSI represents the strain placed on the muscle-tendon unit, our findings indicate that tolerance to the eccentric loading placed on the muscle-tendon unit was not compromised [49]. It would seem strange that we found reduction in stiffness but not RSI when they both represent SSC capability. However, the study by [81] indicated that RSI has a limited amount of common variance with leg stiffness in children. The findings of the current study are not in agreement with a recent study [46] which shows a reduction in RSI immediately post competitive match in soccer players in U14 and U16 players. In addition, other previously mentioned studies on male youth soccer players [38,45] showed a significant reduction in RSI after a competitive soccer game and simulated match-play respectively. Likewise, in the case of leg stiffness, a comparison of differences in the change in RSI between age categories did not demonstrate any age related effects (*p* = 0.130, r = 0.322). 

### 4.7. Limitations

One limitation of this study is that observed indicators were not measured during half time and during other parts of the soccer season, and we did not have the data to explore the reliability of measured outcomes. Moreover, our study actually included players with regular maturation [82,83] and not early maturation typical for elite players. Since it has been shown that maturation status influences the players’ body composition, which is probably the factor influencing the level of fatigue in SAFT [37,84,85], this might highly influence the results. Another limitation could be that the execution of changes of direction during the SAFT^90^ was done by the individual players preferentially and therefore workload on kicking and stance leg could have been different.

## 5. Conclusions

The results of a two-year study on youth male soccer players showed that a simulated soccer protocol (SAFT^90^) induced a significant decrease in most of the observed neuromuscular performance indicators at both ages. Greater changes were found in the older age group (U17); however, most of these differences did not reach statistical significance so we cannot confirm that the effects of a soccer related fatigue protocol on indicators of neuromuscular functions associated with noncontact ACL and H injuries is age dependent. This may be due to the fact that there was only a one-year difference between measurements of the players and that they were post PHV at the first test occasion. With respect to incidence and consequences of ACL and H injuries in soccer, this type of diagnostics has the potential for being effective in developing training strategies to affect neuromuscular mechanisms important for joint stability and reduce injuries in youth soccer, in particular if the results are reflective in flexible planning of the training cycles, especially in competitive microcycle with higher competitive match-play demands and/or cycles with higher level of training and/or competitive demands on players.

## Figures and Tables

**Figure 1 ijerph-17-08579-f001:**
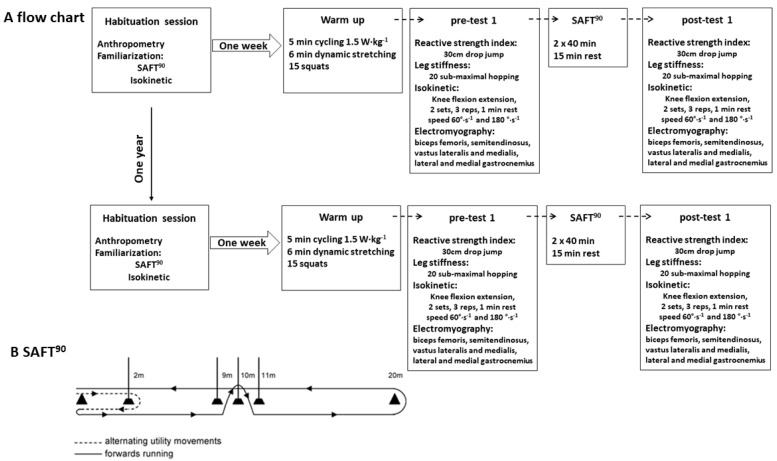
(**A**) Flow chart of performed procedures. (**B**) Diagram of the SAFT^90^ (adopted from Small, McNaughton, Greig, & Lovell, 2010). SAFT^90^ = soccer-specific protocol to induce fatigue.

**Figure 2 ijerph-17-08579-f002:**
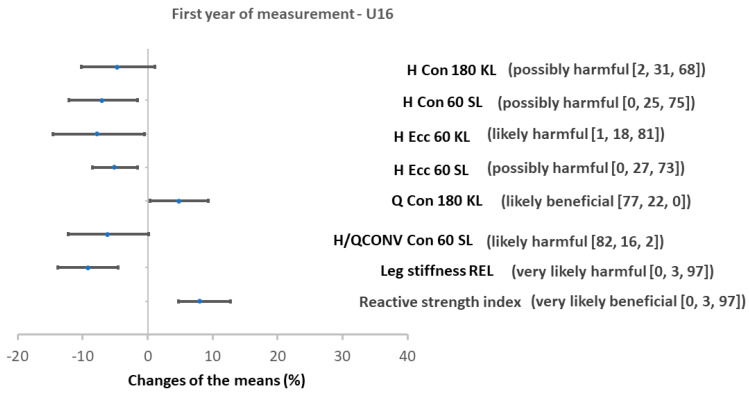
Percentage changes in strength measurement after SAFT^90^ protocol in U16. Values are changes in means with confidence intervals (as %), bracket = (effect [odds ratios]), H = hamstrings, Q = quadriceps, Strength is described by the peak torque, H/Q_CONV_ = conventional ratio of H and Q, SL = stance nondominant lower limb, KL = kicking dominant lower limb.

**Figure 3 ijerph-17-08579-f003:**
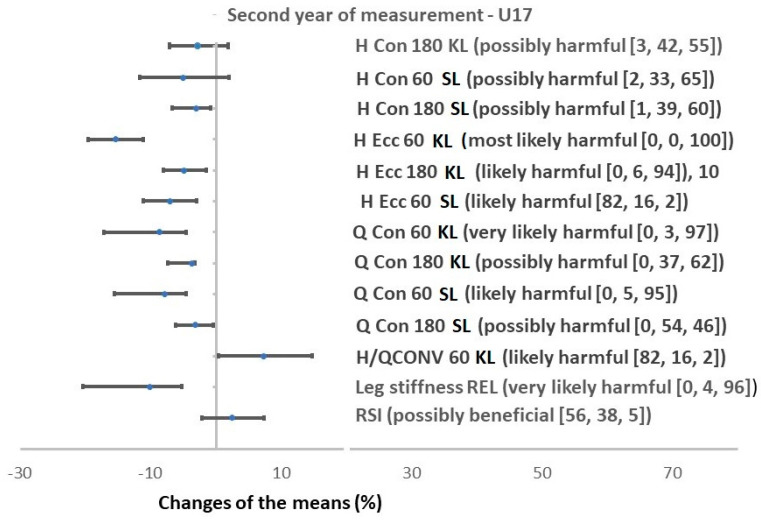
Percentage changes in strength measurement after SAFT^90^ protocol in U17. Values are changes in means with confidence intervals (%), bracket = (effect [odds ratios]), Strength is described by the peak torque, H = hamstrings, Q = quadriceps, H/Q_CONV_ = conventional ratio of H and Q, SL = stance nondominant lower limb, KL = kicking dominant lower limb.

**Figure 4 ijerph-17-08579-f004:**
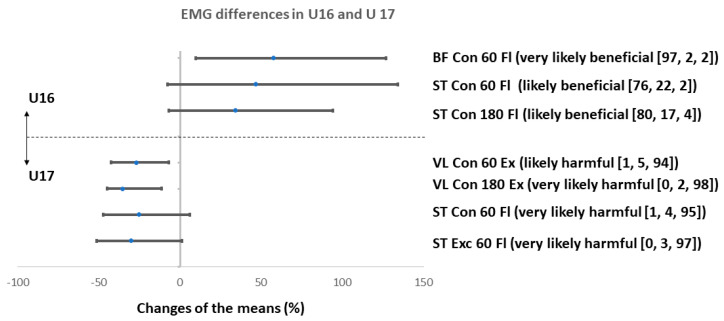
Percentage change in thigh muscle EMG mean frequency after SAFT90 protocol in U16 and U17. Values are changes of means with confidence intervals (as %), bracket = (effect [odds ratios]), BF = biceps femoris, ST = semitendinosus, VL = vastus lateralis quadriceps, Fl = flexion.

**Table 1 ijerph-17-08579-t001:** Physical characteristics of participants (*n* = 11).

Variable	Age (y)	PHV Offset (y)	Stature (cm)	Body Mass (kg)
U16 year 1	16.0 ± 0.4	+2.24 ± 0.71	178.8 ± 6.4	67.5 ± 7.8
U17 year 2	17.0 ± 0.4	+3.31 ± 0.57	180.9 ± 5.7	71.4 ± 6.6

PHV—age from peak height velocity, U16 = first year of testing when the group was under 16 years of calendar age, U17 = second year of testing when the group was under 17 years of calendar age.

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
