# Peer review of "Effect of a Simulated Match on Lower Limb Neuromuscular Performance in Youth Footballers—A Two Year Longitudinal Study"

_ijerph, 2020, doi:10.3390/ijerph17228579_

Round 1
Reviewer 1 Report
Anterior cruciate ligament (ACL) rupture and injury, which are orthopedic injuries in soccer players, as well as hamstring separation, occur more frequently in young soccer players. In this study, the authors hypothesized that the intensity of the game would reduce the strength of the knee flexor muscle group and control of the joint movement.In this study, youth soccer players, who are progressing from U16 to U17 generations, were subjected to SAFT90, which was equivalent to a game, and the muscles acting on the knee flexion movement before and after group (quadriceps: Q, hamstrings: H), and a qualitative reduction in muscle strength and balance between H and Q, as well as a qualitative reduction in muscle discharge and leg stiffness in the knee joint muscle group. The results suggest that ACL and hamstring-related injuries may be due to the loss of muscle strength and other factors over the course of a game.
The results of this study are expected to provide important information for young footballers in order to improve their performance without injury and to improve their physical fitness.
However, the results and discussion presented in this study are insufficient. For example, EMG is measured in muscle strength measurements to show frequency changes in muscle force exertion and muscle discharge, but it is logically developed without a literature review showing how changes in fatigue can be revealed from frequency.
Leg stiffness only reports changes in spring stiffness, but there is no review of the paper on 'why the change in stiffness occurred'. In fact, many previous studies have shown parameters related to leg stiffness adjustment, but no attempt has been made to validate the stiffness adjustment with reference to such studies.
Nor is it indicated how the results of the study that I have presented compare to other previous studies in terms of other parameters.
Introduction
The overall structure of the Introduction is understood to consist of the following six components as,
The first paragraph describes an increase in the frequency of injuries as football players' training time and number of games increases.
The second paragraph introduces a study that reports that the most common injuries are ACL injuries and hamstring injuries, which are also increasing in the youth generation.
The third paragraph describes that the parameters associated with the occurrence of these injuries which are muscle weakness, reduced muscle activity level, reduced leg stiffness, and reduced rebound strength index, as represented by the stretch-shortening cycle.
The fourth paragraph describes the review of recent studies on knee instability caused by anatomical dysfunction of the hamstrings in adults.
The fifth paragraph describes review of studies on injuries that occur in growing adolescent football players and their causes, indicating that they may be involved in hamstring and quadriceps muscle strength imbalances, but there are no consistent research reports.
The sixth paragraph describes that, based on the above literature studies, the authors aimed to find out whether acute effects of a load test that replicates post-match physical fatigue in growing youth soccer players affect ACL and hamstring injuries, as no studies have examined the acute effects of a load test that replicates post-match physical fatigue in youth soccer players.
Although the contents of the description are very important, they only list the facts of the past, and they do not show the readers the problems of each item or the problems and unresolved aspects of the youth. The reason for this may be due to the slightly redundant structure of the text. Also, readers who read this text will have to understand the gist of the text while reading each paragraph, which will be tiring as they read. This is because the second and subsequent paragraphs do not build on what we have described in the previous paragraphs.
In particular, from the third paragraph onwards, the current situation and problems are listed for each inspection item, and these are repeated, making it difficult to grasp the flow of the sentence. Therefore, it would be better to list the items related to ACL and hamstring injuries first, briefly describe the views of previous research on each item, and then, in the following paragraphs, describe what has been found and not yet considered in the research with youth footballers, and then present the issues and link them to the objectives of the study Isn't it?
Method
Line 134 Figure.1
Although the authors have presented the inspection procedures, they only show the general flow, which may not tell the reader what they are doing. Especially, it is good to have the silhouette diagram and actual waveforms (EMG, ground reaction force, isotonic muscular force, etc.) that show the items tested in the Pre-Test and Post-Test. Why don't we display the diagrams of the exercise test performed on the subjects?
From Line 175 to 180 in EMG analysis
The authors explain that the EMG signal is converted into a frequency band by using specialized analysis software, and the change in the level of muscle activity is shown by the average frequency of the EMG signal. It is not stated whether the filter is applied at the cutoff frequency or in the adopted frequency band.
The authors have also performed frequency analysis of EMG signals to compare changes in muscle activity levels before and after the exercise stress test, but have not shown any previous studies that provide evidence of changes in EMG frequency due to muscle fatigue. It should be shown why the authors use this method, as it provides a rationale.
From Line 189 to 200
2.6 Leg stiffness
The authors ordered subjects hopping on a force platform and used the method of Dalleau et al. in the subsequent calculation of leg stiffness.
However, previous research on leg stiffness adjustment has shown that the parameters related to the modulation of leg stiffness are (1) the angle of the joint during landing, (2) the preliminary activity of the muscle prior to landing, (3) whether the landing impact force is utilized as a propulsive force for the next hopping, and (4) the Peak value of vertical ground reaction force, (5) the downward displacement of the body's center of gravity from landing to mid-stance, which equals to shortening of the linear spring of the leg, (6) the slope of the curve consisted by the center of gravity displacement and Fz, (7) ground contact duration, (8) ankle stiffness, etc... The relationship between these parameters and leg stiffness modulation has been reported in many previous studies. These parameters may provide a useful explanation for why the leg stiffness changed after the loading test. Among them, parameters (3) to (7) were obtained by analyzing the data of the force platform, which used in the experiment.
Results
Line 229 There is no ’Supplementary file 2’.
From Line 278 to 282
The authors compare the two ages in the measurement items and describe a significant change in the change from U16 to U17, but they only describe the numbers in the text and are not reader-friendly. It may be necessary to show a figure or table so that the reader can understand it.
Discussion
Line 284 to 291
The authors describe here the main results of their study and discuss the first considerations from it. However, it is assumed that readers may have forgotten the purpose of this study because the text written in methods and results is somewhat redundant. Therefore, it is necessary to reaffirm to the reader the purpose of this study and should be stated again.
Line 289 to 291 Authors state that ‘These results and the comparison of differences in changes of the observed characteristics do not indicate… ’. However, the statement 'and the comparison of differences in changes of the observed characteristics' is redundant and is not necessary.
Line 309 to 314
It is not clear what the authors' results show from what they describe here. It does not convey whether the authors have described it in such a way that it has a connection to the paragraphs above it and the paragraphs below it. Please describe them in a way that conveys the authors' intentions.
Line 338 to 352
The authors discuss the H/Q ratio. However, the reader is unable to read the authors' intentions because they do not describe how their results are positioned in comparison to previous studies and what the situation is.
Line 357 to 359
The authors describe 'These results support the findings of the changes in PTs and ...', but 'the Where does the 'findings of the ...' refer to? The result of the authors? Or is it the result of previous research? This is unclear, and the authors should be clear about it.
Line 371 to 373
It is difficult to ... isokinetic knee action[61]. I don't know what this sentence means. What are the authors trying to tell us? And by comparing their results with the literature [61], what do the authors want to convey to their readers? but the reader cannot understand it.
Line 411 to 432
Both U16 and U17 described a significant decrease in leg stiffness after the exercise stress test.
What do the authors believe is the cause of this decrease in stiffness? Following the results of previous studies examining leg stiffness modulation, the authors should be able to determine whether this is due to a decrease in the maximum value of Fz, increased landing impact force, increased downward displacement of the center of gravity, or increased ground contact time. Based on such previous studies and results, the authors can have a more sophisticated discussion.
As commented in the Methods section, since kinematic and mechanical changes are involved in the modulation of leg stiffness, the parameters obtained from the ground reaction force meter should be presented in the results and discussed together with the results.
Author Response
Anterior cruciate ligament (ACL) rupture and injury, which are orthopedic injuries in soccer players, as well as hamstring separation, occur more frequently in young soccer players. In this study, the authors hypothesized that the intensity of the game would reduce the strength of the knee flexor muscle group and control of the joint movement. In this study, youth soccer players, who are progressing from U16 to U17 generations, were subjected to SAFT90, which was equivalent to a game, and the muscles acting on the knee flexion movement before and after group (quadriceps: Q, hamstrings: H), and a qualitative reduction in muscle strength and balance between H and Q, as well as a qualitative reduction in muscle discharge and leg stiffness in the knee joint muscle group. The results suggest that ACL and hamstring-related injuries may be due to the loss of muscle strength and other factors over the course of a game. The results of this study are expected to provide important information for young footballers in order to improve their performance without injury and to improve their physical fitness. However, the results and discussion presented in this study are insufficient. For example, EMG is measured in muscle strength measurements to show frequency changes in muscle force exertion and muscle discharge, but it is logically developed without a literature review showing how changes in fatigue can be revealed from frequency.
Answer: Thank you for detailed and constructive comments to our manuscript. We have revised the whole manuscript and made a plethora of changes based on reviewers comments. E.g. including the justification of EMG frequency as a measure of muscle fatigue mentioned above by adding following references bellow:
De Luca, C. J. (1997). The Use of Surface Electromyography in Biomechanics, Journal of Applied Biomechanics, 13(2), 135-163.
Thongpanja, S., Phinyomark, A., Phukpattaranont, P., Limsakul, C. (2013). Mean and Median Frequency of EMG Signal to Determine Muscle Force based on Timedependent Power Spectrum. Elektronika ir elektrotechnika, 19(3), 51-56.
Leg stiffness only reports changes in spring stiffness, but there is no review of the paper on 'why the change in stiffness occurred'. In fact, many previous studies have shown parameters related to leg stiffness adjustment, but no attempt has been made to validate the stiffness adjustment with reference to such studies.
Answer: We have suggested some changes based on previous literature around ground contact time. We have added in some discussion around why we think stiffness changes and the implications that this has for both injury risk and performance
Nor is it indicated how the results of the study that I have presented compare to other previous studies in terms of other parameters.
Answer: We have added some greater articulation around how our findings relate to parameters in current literature
Introduction
The overall structure of the Introduction is understood to consist of the following six components as,
The first paragraph describes an increase in the frequency of injuries as football players' training time and number of games increases.
The second paragraph introduces a study that reports that the most common injuries are ACL injuries and hamstring injuries, which are also increasing in the youth generation.
The third paragraph describes that the parameters associated with the occurrence of these injuries which are muscle weakness, reduced muscle activity level, reduced leg stiffness, and reduced rebound strength index, as represented by the stretch-shortening cycle.
The fourth paragraph describes the review of recent studies on knee instability caused by anatomical dysfunction of the hamstrings in adults.
The fifth paragraph describes review of studies on injuries that occur in growing adolescent football players and their causes, indicating that they may be involved in hamstring and quadriceps muscle strength imbalances, but there are no consistent research reports.
The sixth paragraph describes that, based on the above literature studies, the authors aimed to find out whether acute effects of a load test that replicates post-match physical fatigue in growing youth soccer players affect ACL and hamstring injuries, as no studies have examined the acute effects of a load test that replicates post-match physical fatigue in youth soccer players.
Although the contents of the description are very important, they only list the facts of the past, and they do not show the readers the problems of each item or the problems and unresolved aspects of the youth. The reason for this may be due to the slightly redundant structure of the text. Also, readers who read this text will have to understand the gist of the text while reading each paragraph, which will be tiring as they read. This is because the second and subsequent paragraphs do not build on what we have described in the previous paragraphs.
In particular, from the third paragraph onwards, the current situation and problems are listed for each inspection item, and these are repeated, making it difficult to grasp the flow of the sentence. Therefore, it would be better to list the items related to ACL and hamstring injuries first, briefly describe the views of previous research on each item, and then, in the following paragraphs, describe what has been found and not yet considered in the research with youth footballers, and then present the issues and link them to the objectives of the study Isn't it?
Answer:
Thank you for suggested changes in the structure of the introduction. We agree with it and we followed it when making changes in this section (as well as recommendations of the 3rd reviewer). Specifically, we change the flow to:
1- Increase in the frequency of injuries in youth soccer, influence of local fatigue on factors associated with lower limb non-contact injuries, including neuromuscular performance parameters
- Previous research view on used neuromuscular parameters, their association with ACL and hamstring injuries (+ also soccer performance) and findings of studies focused on post-match or simulated match-play changes in observed neuromuscular parameters in youth soccer players.
- What has not been found and not yet considered in the research in this topic, followed by the aim of the study.
Method
Line 134 Figure.1
Although the authors have presented the inspection procedures, they only show the general flow, which may not tell the reader what they are doing. Especially, it is good to have the silhouette diagram and actual waveforms (EMG, ground reaction force, isotonic muscular force, etc.) that show the items tested in the Pre-Test and Post-Test. Why don't we display the diagrams of the exercise test performed on the subjects?
Answer: We have put list of whole measurements design in the figure 1. On the other hand, we are actually using the quantitative values not the curves or gradients. Therefore, the diagrams of waveform might be misleading.
From Line 175 to 180 in EMG analysis
The authors explain that the EMG signal is converted into a frequency band by using specialized analysis software, and the change in the level of muscle activity is shown by the average frequency of the EMG signal. It is not stated whether the filter is applied at the cutoff frequency or in the adopted frequency band.
The authors have also performed frequency analysis of EMG signals to compare changes in muscle activity levels before and after the exercise stress test, but have not shown any previous studies that provide evidence of changes in EMG frequency due to muscle fatigue. It should be shown why the authors use this method, as it provides a rationale.
Answer: We have added corresponding information to the introduction section including references (mentioned above within the answer to the 1st comment).
From Line 189 to 200
2.6 Leg stiffness
The authors ordered subjects hopping on a force platform and used the method of Dalleau et al. in the subsequent calculation of leg stiffness.
However, previous research on leg stiffness adjustment has shown that the parameters related to the modulation of leg stiffness are (1) the angle of the joint during landing, (2) the preliminary activity of the muscle prior to landing, (3) whether the landing impact force is utilized as a propulsive force for the next hopping, and (4) the Peak value of vertical ground reaction force, (5) the downward displacement of the body's center of gravity from landing to mid-stance, which equals to shortening of the linear spring of the leg, (6) the slope of the curve consisted by the center of gravity displacement and Fz, (7) ground contact duration, (8) ankle stiffness, etc... The relationship between these parameters and leg stiffness modulation has been reported in many previous studies. These parameters may provide a useful explanation for why the leg stiffness changed after the loading test. Among them, parameters (3) to (7) were obtained by analyzing the data of the force platform, which used in the experiment.
Answer: We fully understand the point that the reviewer is making here, however it is important to note that this study does not soley focus on limb stiffness but also EMG analysis and isokinetic outcomes. If this were a study focusing exclusively on limb stiffness, we might look at a multitude of parameters relating to that outcome variable. Moreover, we believe that our parameters selection represent well our aim. We have however, added in some discussion around potential changes in ground contact time and subsequent changes in centre of mass displacement.
Results
Line 229 There is no ’Supplementary file 2’.
Answer: Sorry for this mistake, we have uploaded the raw data file now
From Line 278 to 282
The authors compare the two ages in the measurement items and describe a significant change in the change from U16 to U17, but they only describe the numbers in the text and are not reader-friendly. It may be necessary to show a figure or table so that the reader can understand it.
Answer: We have only found age related differences in a few of our EMG outcome measures and as a main part of our conclusions suggest that there are not age-related differences in the neuromuscular response. We therefore do not think it is appropriate to show these few age differences graphically and thus present them numerically in the manuscript.
Discussion
Line 284 to 291
The authors describe here the main results of their study and discuss the first considerations from it. However, it is assumed that readers may have forgotten the purpose of this study because the text written in methods and results is somewhat redundant. Therefore, it is necessary to reaffirm to the reader the purpose of this study and should be stated again.
Answer: We added the purpose of the study as recommended.
Line 289 to 291 Authors state that ‘These results and the comparison of differences in changes of the observed characteristics do not indicate… ’. However, the statement 'and the comparison of differences in changes of the observed characteristics' is redundant and is not necessary.
Answer: We agree with the reviewer that this part of the sentence is redundant and we deleted it.
Line 309 to 314
It is not clear what the authors' results show from what they describe here. It does not convey whether the authors have described it in such a way that it has a connection to the paragraphs above it and the paragraphs below it. Please describe them in a way that conveys the authors' intentions.
Answer: In the preceding paragraph, we highlight that the main effects of the soccer task affect predominantly the hamstrings when working eccentrically. This paragraph notes that our findings are consistent with the extant literature. We have taken out the sentence that relates to isometric data for clarity around eccentric actions
Line 338 to 352
The authors discuss the H/Q ratio. However, the reader is unable to read the authors' intentions because they do not describe how their results are positioned in comparison to previous studies and what the situation is.
Answer: We do not fully share reviewers' view that we did not describe how the results are positioned in comparison to previous studies. In the discussion, we stated that our results support findings of two previous studies on youth soccer players, however contradict the results of study on adult players. Nevertheless, we added new information and reference to make this part of the discussion more transparent.
Line 357 to 359
The authors describe 'These results support the findings of the changes in PTs and ...', but 'the Where does the 'findings of the ...' refer to? The result of the authors? Or is it the result of previous research? This is unclear, and the authors should be clear about it.
Answer: We have reworded this sentence so that it is clear that we mean that the results in the case of H/Q ratios support changes in PTs in our study.
Line 371 to 373
It is difficult to ... isokinetic knee action[61]. I don't know what this sentence means. What are the authors trying to tell us? And by comparing their results with the literature [61], what do the authors want to convey to their readers? but the reader cannot understand it.
Answer: We have removed this sentence as it was confusing and the following paragraph provides a potential explanation of our findings
Line 411 to 432
Both U16 and U17 described a significant decrease in leg stiffness after the exercise stress test. What do the authors believe is the cause of this decrease in stiffness? Following the results of previous studies examining leg stiffness modulation, the authors should be able to determine whether this is due to a decrease in the maximum value of Fz, increased landing impact force, increased downward displacement of the center of gravity, or increased ground contact time. Based on such previous studies and results, the authors can have a more sophisticated discussion.
Answer: We have added in some discussion around potential changes in ground contact time and subsequent changes in centre of mass displacement. We have also dicussed the possible changes in pre-activation prior to ground contact
As commented in the Methods section, since kinematic and mechanical changes are involved in the modulation of leg stiffness, the parameters obtained from the ground reaction force meter should be presented in the results and discussed together with the results.
Answer: See previous comments regarding the leg stiffness data and potential explanation of our findings.
Reviewer 2 Report
The topic of the present manuscript is interesting and the results will increase the body of knowledge in this area. However, the experimental design is not easy to follow and the results should be interpreted with caution. I suggest the following considerations:
Page 1, line 2. In the title the term “Acute effect” appears conflictual with the term “longitudinal study”. The Authors should remove “acute effect” or modify the title.
Page 1, lines 15-17. The sentence should be removed as the Authors did not determine a cause-effect relationship between a simulated soccer match play and ACL-hamstrings injury risk factors.
Page 1, line 20. The acronyms “SAFT90 should be explained in the first citation.
Page 1, lines 25-26. The sentence: “This study demonstrates an increase in injury risk…..should be removed as the study did not investigate a cause-effect relationship (see my previous comment). The Authors should reformulate the conclusions focusing on their main results.
Page 1, lines 29-30. Please add another impacted reference to reinforce the statement.
Pages 1-2, lines 29-95. This section is well written and include appropriate references, but in my opinion should be shortened.
Page 3, lines 96-104. This paragraph should be rewritten. The present study did not explore in which way SAFT90 increased ACL and hamstrings injuries risk factors in youth athletes! The study analysed the acute effect induced by SAFT90 on the neuromuscular system. The associations between ACL and hamstrings injuries risk factors is a speculative sentence that could be argued in the discussion…
Page 3, line 107. Please, include the country.
Page 3, lines 107-127. This section appears confusing for the reader. Some information regarding the participants could be included in the table 1 (i.e. the number of participants in the year 1 and year 2).
Page 3, lines 122-123. The statement is not clear. Was the study approved by the ethical committee? Please, specify in the text.
Page 4, Figure 1. The figure 1 should be modified, it is not clear. Please, include a figure that shows the experimental design.
Page 4, line 145-isokinetic dynamometry. The Authors should explain (in the text) why they used this procedure. What is the rationale? The isokinetic dynamometry is performed at constant velocity while the neuromuscular system of a soccer player works continuously against the gravity (i.e. accelerations, jumps)!! Additionally, the participants performed an open kinetic chain exercise while the soccer player performs closed kinetic chain exercises during their habitual movements (with some exceptions, i.e. kicking a ball).
Page 4, line 148. What does “gravitational correction” means? Please, explain in the text.
Page 4, lines 161, 164-166. The Authors should report the ICC calculated in their measurements.
Page 4, lines 166-167, 172-175. Was the EMG recorded by using different electrodes? Why did you used two different electrodes? Please, explain in the text. Did the Author follow the SENIAM or ISEK recommendations? If yes, please, include reference.
Page 5, line 198. Please, correct the citation Dalleau et al. [62].
Page 8-12, discussion. Overall, this section should be partially rewritten. The Authors reported the results (the p-values should be reported in the section results) but failed to explain them and expand their physiological reasoning taking in account the training regimen performed by the players in the two years in relation to their growth.
Page 8, lines 285-286. Please, explain the sentence better. What does “irrespective of chronological age” mean?
Page 8, line 287. Please, use U16 and U17 in place of older and younger.
Page 10, lines 374-376. The Authors should explain in physiological terms the changes in electromechanical delay following SAFT90 .
Page 11, lines 447-449. The sentence is not clear.
Author Response
The topic of the present manuscript is interesting and the results will increase the body of knowledge in this area. However, the experimental design is not easy to follow and the results should be interpreted with caution. I suggest the following considerations:
Answer: Thank you for your comments and time spent on improving our manuscript. We put special focus to making the experimental design clear and on adequate interpretation of our results.
Page 1, line 2. In the title the term “Acute effect” appears conflictual with the term “longitudinal study”. The Authors should remove “acute effect” or modify the title.
Answer: We agree that the acute might be confusing here, therefore we deleted the “Acute”.
Page 1, lines 15-17. The sentence should be removed as the Authors did not determine a cause-effect relationship between a simulated soccer match play and ACL-hamstrings injury risk factors.
Answer: We have revised it and now we are claiming “neuromuscular performance“ only.
Page 1, line 20. The acronyms “SAFT90 should be explained in the first citation.
Answer: done
Page 1, lines 25-26. The sentence: “This study demonstrates an increase in injury risk…..should be removed as the study did not investigate a cause-effect relationship (see my previous comment). The Authors should reformulate the conclusions focusing on their main results.
Answer: We change the conclusion to “neuromuscular performance“ only.
Page 1, lines 29-30. Please add another impacted reference to reinforce the statement.
Answer: Wee added one more relevant reference: Stølen, T., Chamari, K., Castagna, C., & Wisløff, U. (2005).
Pages 1-2, lines 29-95. This section is well written and include appropriate references, but in my opinion should be shortened.
Answer: We deleted some sentences in the introduction. However, at the same time we accepted comments of the 1st reviewer, and also the 3rd reviewer and added some new information, especially about the relation of fatigue, injury and neuromuscular performance parameters used in our study.
Page 3, lines 96-104. This paragraph should be rewritten. The present study did not explore in which way SAFT90 increased ACL and hamstrings injuries risk factors in youth athletes! The study analysed the acute effect induced by SAFT90 on the neuromuscular system. The associations between ACL and hamstrings injuries risk factors is a speculative sentence that could be argued in the discussion…
Answer: We agree, therefore we removed the speculative sentences.
Page 3, line 107. Please, include the country.
Answer: Included.
Page 3, lines 107-127. This section appears confusing for the reader. Some information regarding the participants could be included in the table 1 (i.e. the number of participants in the year 1 and year 2).
Answer: Thank for this point, the table 1 is showing only participant who finish both years of the study. We now re-write the initial participant statement, where we putt clearly number of recruited and included participants. Thus we change the description and left the table how it is.
Page 3, lines 122-123. The statement is not clear. Was the study approved by the ethical committee? Please, specify in the text.
Answer: Yes, we make this statement more clear including institution name.
Page 4, Figure 1. The figure 1 should be modified, it is not clear. Please, include a figure that shows the experimental design.
Answer: We have included the measurement session design in the Figure 1.
Page 4, line 145-isokinetic dynamometry. The Authors should explain (in the text) why they used this procedure. What is the rationale? The isokinetic dynamometry is performed at constant velocity while the neuromuscular system of a soccer player works continuously against the gravity (i.e. accelerations, jumps)!! Additionally, the participants performed an open kinetic chain exercise while the soccer player performs closed kinetic chain exercises during their habitual movements (with some exceptions, i.e. kicking a ball).
Answer: We put this rationale to the introduction, with appropriate references. Isokinetic strength has been suggested as one of the possible injury predictors (this has been already referenced in the manuscript) and it also correlate with soccer sprints and agility (see added references below).
Booysen MJ, West N, Constantinou D. P-85 The relationships of eccentric and concentric isokinetic strength with sprinting speed in male sub-elite footballers: BMJ Publishing Group Ltd and British Association of Sport and Exercise Medicine, 2016.
Cotte T, Chatard J. Isokinetic strength and sprint times in English premier league football players. Biology of Sport 2011, 28(2):89.
Harper D, Jordan A, Kiely J. Relationships between eccentric and concentric knee strength capacities and maximal linear deceleration ability in male academy soccer players. Journal of strength and conditioning research, 2018
Page 4, line 148. What does “gravitational correction” means? Please, explain in the text.
Answer: It means gravitational correction for the mass of the measured lower limb, which is now specified in the text.
Page 4, lines 161, 164-166. The Authors should report the ICC calculated in their measurements.
Answer: The reproducibility for the IsoMed 2000 dynamometer in measuring concentric and eccentric knee extension has been reported as being high (see the reference in the manuscript).
Page 4, lines 166-167, 172-175. Was the EMG recorded by using different electrodes? Why did you used two different electrodes? Please, explain in the text. Did the Author follow the SENIAM or ISEK recommendations? If yes, please, include reference.
Answer: We did not used different electrodes, the “Ag/AgCl” is just shortcut for the type of electrodes described above in manuscript. Sorry for this inaccuracy, we now deleted the redundant abbreviation. To be fair, A new electrode was used only in the case the original one became loose or fell due to sweating. We have newly nailed it down in the manuscript and added reference bellow:
Konrad, P. (2005). ABC of EMG – A practical introduction to kinesiological electromyography. Retrieved 27. 4. 2005 from the World Wide Web: http://www.noraxon.com/downloads/educational.php3.
Page 5, line 198. Please, correct the citation Dalleau et al. [62].
Answer: Done as suggested.
Page 8-12, Discussion. Overall, this section should be partially rewritten. The Authors reported the results (the p-values should be reported in the section results) but failed to explain them and expand their physiological reasoning taking in account the training regimen performed by the players in the two years in relation to their growth.
Answer: We have reworked the discussion and the potential changes in training across age ranges does not seem to have influenced the fatigue related outcomes as we did not find any age related differences. The result section actually includes the significant p values in the supplementary material, which we believe is not necessary to discussed due to the clarity and number of results, for probability we report alfa level.
Page 8, lines 285-286. Please, explain the sentence better. What does “irrespective of chronological age” mean?
Answer: we meant participant age between both measures. We revised this sentence.
Page 8, line 287. Please, use U16 and U17 in place of older and younger.
Answer: Done as suggested
Page 10, lines 374-376. The Authors should explain in physiological terms the changes in electromechanical delay following SAFT90 .
Answer: As this comment refers to another study and that we have not measured EMD in our study, we do not think it appropriate to detail the physiological factors that make up EMD (so as not to confuse the reader with frequency analysis). We have retained the focus on our outcome measure of EMG frequency analysis.
Page 11, lines 447-449. The sentence is not clear.
Answer: We make reworded this sentence to make it clear.
Reviewer 3 Report
Dear Authors,
I have read your manuscript and I can tell the idea behind the study is interesting. However, the article may benefit from some reorganization and elaboration.
Below you will find a detailed description of the minor/major changes requested:
The introduction needs to be clear what the practical question is that you are trying to address. How the answer to this question is important to the field as this is not clear or obvious? How is this study and impactful study and not trivial as this needs more clarity as well? The key issue here is to make sure you set up your approach to the problem. The approach to the problem is essential in determining and describing the rationale for the study. You have not given a basic rationale for the choices made for the variables used in the study. Please treat to improve this part of the Introduction section.
Statistical analysis and methods in general are well explained. However, the results of a power analysis for the determination of the sample size should be included. If this was not done a priori, the effect size could be used to determine posteriori if the sample size is suitable.
What is the reason for measuring and show peak height velocity in your sample? It seems that it was only done to describe the participants. At this point, it would be important to point out that your results are different from those shown for young elite soccer players. In fact, in your sample the maturity offset is reached at an average age of 14 years, so the adolescent can be classified as "on time maturing". On the other hand, 2 recent studies published in IJERPH show how young elite soccer players are predominantly subject to early maturation. This is a very important topic and therefore this data should be highlighted. It could be that the elite teams select subjects with larger body dimensions and higher performances, characteristics of subjects in advance of maturation and tendentially born in the first trimester (relative-age effect). My suggestion is to include this information and comparisons in a dedicated part in the text. The two useful studies in this regard are:
1) Campa et al. 2019. The Role of Somatic Maturation on Bioimpedance Patterns and Body Composition in Male Elite Youth Soccer Players. Int J Environ Res Public Health. 16(23): 4711. doi: 10.3390/ijerph16234711
2) Toselli et al. 2020. Maturity Related Differences in Body Composition Assessed by Classic and Specific Bioimpedance Vector Analysis among Male Elite Youth Soccer Players. Int J Environ Res Public Health. doi: 10.3390/ijerph17030729
A "take home message" section should be included, as the results shown are many and well crafted. Therefore, the main practical applications should be highlighted.
Furthermore, the authors are urged to explore the MDPI literature and identify which recent studies may be useful in updating the proposed references and hence the discussion section.
Lastly, the manuscript is not very fluid in some parts, therefore a revision of the English is recommended to improve its readability.
Author Response
Dear Authors, I have read your manuscript and I can tell the idea behind the study is interesting. However, the article may benefit from some reorganization and elaboration. Below you will find a detailed description of the minor/major changes requested:
Answer: Thank you for your time spent on improving our manuscript. We tried to clear out the ideas presented in the manuscript and include your suggestions.
The introduction needs to be clear what the practical question is that you are trying to address. How the answer to this question is important to the field as this is not clear or obvious? How is this study and impactful study and not trivial as this needs more clarity as well? The key issue here is to make sure you set up your approach to the problem. The approach to the problem is essential in determining and describing the rationale for the study. You have not given a basic rationale for the choices made for the variables used in the study. Please treat to improve this part of the Introduction section.
Answer: We made changes in particular in the structure of the introduction to justify our research more clearly. To indicate the practical questions the study is addressing, in the 1st paragraph of the introduction, we mentioned that the incidence of lower limb injuries has increased in youth soccer and that the highest incidence of injuries was found at the end of both the first and the second halves of match-play when fatigue is likely present. To set up approach to the problem, we mentioned that local fatigue, as the result of match play workload, is one of the main aetiological factors, which contributes to lower limb non-contact injuries in soccer. We also presented the rationale for the choices made for the variables (saying that fatigue alters muscular activation and co-activation, lower limb kinematics, reactive strength, muscle stiffness and other factors associated with injuries and thus increases the risk of injuries, and also performance). At the end of the introduction, we mentioned how our research may be useful for practice as well.
Statistical analysis and methods in general are well explained. However, the results of a power analysis for the determination of the sample size should be included. If this was not done a priori, the effect size could be used to determine posteriori if the sample size is suitable.
Answer: Since it is a two year study with a drop off in the participant sample, we adjusted the statistical approach for our sample size. We using also reference in the statistical section, where our approach is acceptable for n 11.
Hopkins, W., Marshall, S., Batterham, A., & Hanin, J. (2009). Progressive statistics for studies in sports medicine and exercise science. Medicine and Science in Sports and Exercise, 41(1), 3.
https://www.sportsci.org/2020/MBDss.htm#_ENREF_1
What is the reason for measuring and show peak height velocity in your sample? It seems that it was only done to describe the participants. At this point, it would be important to point out that your results are different from those shown for young elite soccer players. In fact, in your sample the maturity offset is reached at an average age of 14 years, so the adolescent can be classified as "on time maturing".
Answer: Thank you for this specification important for this age group. We have added it to participants description in the section 2.1. We also pay attention to this feature of the observed group in the discussion later.
On the other hand, 2 recent studies published in IJERPH show how young elite soccer players are predominantly subject to early maturation. This is a very important topic and therefore this data should be highlighted. It could be that the elite teams select subjects with larger body dimensions and higher performances, characteristics of subjects in advance of maturation and tendentially born in the first trimester (relative-age effect). My suggestion is to include this information and comparisons in a dedicated part in the text. The two useful studies in this regard are:
1) Campa et al. 2019. The Role of Somatic Maturation on Bioimpedance Patterns and Body Composition in Male Elite Youth Soccer Players. Int J Environ Res Public Health. 16(23): 4711. doi: 10.3390/ijerph16234711
2) Toselli et al. 2020. Maturity Related Differences in Body Composition Assessed by Classic and Specific Bioimpedance Vector Analysis among Male Elite Youth Soccer Players. Int J Environ Res Public Health. doi: 10.3390/ijerph17030729
Answer: Thank you for this suggestion, we have added this topic to the limitation part with suggested references.
A "take home message" section should be included, as the results shown are many and well crafted. Therefore, the main practical applications should be highlighted.
Answer: We have added a few practical applications to the conclusions.
Furthermore, the authors are urged to explore the MDPI literature and identify which recent studies may be useful in updating the proposed references and hence the discussion section.
Answer: We have done the additional literature search and updated introduction and discussion.
Lastly, the manuscript is not very fluid in some parts, therefore a revision of the English is recomended to improve its readability.
Answer: Thank you for checking, a native speaker has checked the grammar, and also the language flow.
Round 2
Reviewer 1 Report
The authors responded to each reviewer's suggestions and proofread the English text. This makes the text much easier to read. In fact, I was able to read the manuscript five times faster than the previous one.
In particular, the Introduction begins with a description of the current state of the research the authors wish to address, followed by a review of studies that have examined the causes of ACL and hamstring injuries associated with increased fatigue, a problem the authors are aware of. Based on this, the authors introduce the studies that show the occurrence and causes of these injuries in the youth generation, and finally, they present the problems based on the present situation. Thus, the reader will have a clear understanding of the aim of this study.
On the other hand, there are a few passages in the text and figures described after the method that could be made a little more reader-friendly, and the authors should revise the text and figures based on their content.
[Figure 1]
In the method section, the content is carefully explained with diagrams, which makes it much easier to understand. However, some readers may be new to the SAFT90 protocol after reading the text in Section 2.7. For them, it may be difficult to imagine what the protocol is. Therefore, to make it easier for the reader to understand, authors should show the diagram of SAFT90 as Figure 1b, based on the diagram of the protocol described in reference number [35]. I think the addition of this figure is necessary to improve the reproducibility and reliability of the study.
[Supplementary File 2]
The data in Supplementary File 2 contains the names of individuals. This could lead to the identification of personal information, which could be detrimental to the individual's future activities as a professional player. For this reason, it is preferable to use the notation Subject 1, 2, 3, ...n, or to change it to Subject a, b, c, d,....x, so that the individual cannot be identified.
[Minor Revision]
Line from 89 to 91 & 92
The abbreviations H and Q are shown abruptly, while the meaning of these abbreviations is described in line 92. The authors should describe the meanings of abbreviations in line 90 and line 92 should be shown in abbreviations only.
Line 99 '(Schultz et al., 1999)'<- This reference should be indicated by the reference number if it is in the reference list.
Line 434 ‘ (Small et al., 2010)’<- This reference should be indicated by the reference number if it is in the reference list.
Line 705 '(Ford et al., 2010)'<- This reference should also be indicated by the reference number.
Author Response
The authors responded to each reviewer's suggestions and proofread the English text. This makes the text much easier to read. In fact, I was able to read the manuscript five times faster than the previous one. In particular, the Introduction begins with a description of the current state of the research the authors wish to address, followed by a review of studies that have examined the causes of ACL and hamstring injuries associated with increased fatigue, a problem the authors are aware of. Based on this, the authors introduce the studies that show the occurrence and causes of these injuries in the youth generation, and finally, they present the problems based on the present situation. Thus, the reader will have a clear understanding of the aim of this study.
Answer: Thank you for this positive feedback. We did our best to improve the quality of this section.
On the other hand, there are a few passages in the text and figures described after the method that could be made a little more reader-friendly, and the authors should revise the text and figures based on their content.
Answer: We tried to make these parts clearer as suggested.
[Figure 1]
In the method section, the content is carefully explained with diagrams, which makes it much easier to understand. However, some readers may be new to the SAFT90 protocol after reading the text in Section 2.7. For them, it may be difficult to imagine what the protocol is. Therefore, to make it easier for the reader to understand, authors should show the diafram of SAFT90 as Figure 1b, based on the diagram of the protocol described in reference number [35]. I think the addition of this figure is necessary to improve the reproducibility and reliability of the study.
Answer: We have added the diagram of SAFT90 protocol as recommended.
[Supplementary File 2]
The data in Supplementary File 2 contains the names of individuals. This could lead to the identification of personal information, which could be detrimental to the individual's future activities as a professional player. For this reason, it is preferable to use the notation Subject 1, 2, 3, ...n, or to change it to Subject a, b, c, d,....x, so that the individual cannot be identified.
Answer: The supplementary material was blinded, thank you for point to this detail.
[Minor Revision]
Line from 89 to 91 & 92
The abbreviations H and Q are shown abruptly, while the meaning of these abbreviations is described in line 92. The authors should describe the meanings of abbreviations in line 90 and line 92 should be shown in abbreviations only.
Answer: We have described the abbreviations first time, when they appeared.
Line 99 '(Schultz et al., 1999)'<- This reference should be indicated by the reference number if it is in the reference list.
Answer: Done as suggested.
Line 434 ‘ (Small et al., 2010)’<- This reference should be indicated by the reference number if it is in the reference list.
Answer: Done as suggested.
Line 705 '(Ford et al., 2010)'<- This reference should also be indicated by the reference number.
Answer: Done as suggested.
Reviewer 2 Report
The Authors revised the manuscript following the suggestions of the present reviewer. The Authors have made a great job and the manuscript has been improved considerably. I have just few minor points:
Page 2, lines 63-69. Please, check the sentence and the references.
Page 2, lines 75-76. The reference number is lacking.
Page 3, lines 106-107. Stiffness is not an “example of a neuromuscular feed-forward mechanism”. Leg stiffness adjustments are also based on the feed-forward mechanism. Please, correct the sentence.
Page lines 112-113. Please, do not use capital letters for the term “reactive strength index”.
Page 3, 120-121. Please, check the sentence.
Page 3, lines 136-137. Please, check the sentence.
Page 4, lines 157-161. Please, consider to remove the sentence and include the number of players in the table 1.
Page 5, figure 1. Please, correct the terms pre-test e post-test.
Page 5, lines 213-214. It is well known that the reliability of the isokinetic measurements are high! However, the Authors usually report the ICC values of the measured variables in their own study. Therefore, if you are not able to assess the ICC of the measured variables (i.e. isokinetic, leg stiffness, RSI and EMG), you should include this issue in the limitations.
Page 9, line 342. Please, consider to not use acronyms in the discussion (i.e. PT, KL).
Page 11, lines 430-431. Please, check the reference.
The p-values are sometimes reported with capital letters. Please, check all over the manuscript.
Author Response
The Authors revised the manuscript following the suggestions of the present reviewer. The Authors have made a great job and the manuscript has been improved considerably. I have just few minor points:
Answer: Thank you for this positive feedback. We did our best to improve the quality of the manuscript according to your comments. We also tried to address the suggestions below.
Page 2, lines 63-69. Please, check the sentence and the references.
Answer: We simplified the sentence and double-checked the references, which were confusing due to the reference manager formatting.
Page 2, lines 75-76. The reference number is lacking.
Answer: We have added the reference number.
Page 3, lines 106-107. Stiffness is not an “example of a neuromuscular feed-forward mechanism”. Leg stiffness adjustments are also based on the feed-forward mechanism. Please, correct the sentence.
Answer: We have corrected this statement according to your suggestion.
Page lines 112-113. Please, do not use capital letters for the term “reactive strength index”.
Answer: Done as suggested.
Page 3, 120-121. Please, check the sentence.
Answer: This error was just in the track changes mode. We revised and it is clear.
Page 3, lines 136-137. Please, check the sentence.
Answer: This error was just in the track changes mode. We revised and it is clear.
Page 4, lines 157-161. Please, consider to remove the sentence and include the number of players in the table 1.
Answer: This text description has been requested in previous review round. Therefore, we left it in the text form (There was again confusing track changes double text, which we corrected).
Page 5, figure 1. Please, correct the terms pre-test e post-test.
Answer: Done
Page 5, lines 213-214. It is well known that the reliability of the isokinetic measurements are high! However, the Authors usually report the ICC values of the measured variables in their own study. Therefore, if you are not able to assess the ICC of the measured variables (i.e. isokinetic, leg stiffness, RSI and EMG), you should include this issue in the limitations.
Answer: We included this into limitations as suggested.
Page 9, line 342. Please, consider to not use acronyms in the discussion (i.e. PT, KL).
Answer: Thank you for this suggestion. We decided not to use acronym in the cases of kicking and supporting legs only as we suppose that the acronyme PT is more commonly used in such studies. To be homogenous we followed this rule in all the sections of the manuscript.
Page 11, lines 430-431. Please, check the reference.
Answer: This error was just in the track changes mode. We revised and it and is clear now.
The p-values are sometimes reported with capital letters. Please, check all over the manuscript.
Answer: Thank you for this notice, now we use only “p” in the main text and supplementary materials.
Reviewer 3 Report
The Author addressed all the suggestions. In this form the manuscript is acceptable for the publication.
Author Response
The Author addressed all the suggestions. In this form the manuscript is acceptable for the publication.
Answer: Thank you for your positive evaluation, We have tried our best to improve the manuscript.
This manuscript is a resubmission of an earlier submission. The following is a list of the peer review reports and author responses from that submission.
Round 1
Reviewer 1 Report
This manuscript submitted by Lehnert and co-workers seems like a draft version of an original article. While I understand that authors are not native English, a revision of the manuscript is fundamental to improve its readability.
The abstract presents acronyms without ever being preceded by full names. Also in this section no statistical data was reported.
The introduction needs to be clear what the practical question is that you are trying to address. How the answer to this question is important to the field as this is not clear or obvious? How is this study and impactful study and not trivial as this needs more clarity as well. The key issue here is to make sure you set up your approach to the problem. The approach to the problem is essential in determining and describing the rationale for the study. You have not given a basic rationale for the choices made for the variables used in the study.
The statistics are very confusing without a clear experimental design and an a priori analysis that justifies the sample size.
The manuscript largely lacks illustrative representation by summarizing correlations in tables without showing the original data but rather an r/r2/p value. This is a poor representation of scientific data.
The discussion section is very descriptive and offers limited comparisons to previous research. Similarly, how do practitioner benefit from that? Again, the discussion section fails to relate the findings to this particular application of interest. Authors are therefore encouraged to make substantial changes throughout to improve the overall quality. In the current form the rationale for the study is not clear, the new value is unclear, and I have difficulties finding specific take home messages for practitioners.
Finally, the authors did not bother to observe the guidelines required by the IJERPH. This should be done carefully when submitting an article to a Journal that requests and accepts only high quality papers.
Reviewer 2 Report
I am grateful for the opportunity to review the manuscript presented to me. I believe the paper is worth considering for publication.
Here are some more specific comments:
Page 3, line 76, it must include the duration of the training sessions
Page 3, line 81, it must include the name and the number of Ethics Committee authorization
The article's conclusion does not highlight the great problem of low statistical significance. I think you need to look more into the reasons and the issues that led to this result.
Reviewer 3 Report
The introduction section should be more focused and better organized and elaborated. Fatigue should be defined in the introduction. It is not clear to what type of fatigue are the authors referring to local or systemic. It is not clear whether the risk of injury is higher in children or adults. Moreover, injury incidence data should be reported. The methods used are poorly described, and some interpreted wrongly. The statistical analysis is entirely unacceptable (see more below). The literature does not well support the inference made from uncertain changes in H/Q and injury risk. The neurophysiological assessments performed are rather elementary and not well described. Therefore, the conclusions achieved are based on Bayesian estimations (wrong), applying methods with low reliability. The latter is further complicated by the lack of a control group, the small size of the sample population and the small number of assessments performed longitudinally.
Major:
There are too many English grammar issues (as just mentioned a few, there are many more). This is unacceptable.
L35. What means “neuromuscular control interacts in ACL load.”
L36: What means “negatively influences muscular co-activation” Who do you know the influence is negative? What means "negatively influences" (it increases or decreases what function?).
L55. What is “PT”
L56. What is muscular stability: I knew about joint stability but muscular? Moreover, “muscular stability” was not measured in ref 41.
L61-62. Actually, there is considerable research in this field.
Methods
More information regarding the players are needed (training volume, previous injuries, fitness level, training history, etc.).
Report the reproducibility of measurements.
How was the gravitational correction performed?
The reliability of the H/Q ratios is very low and limits the detection capacity of any effect on this variable.
The information regarding the isokinetic and EMG analysis is incomplete. Averaging interval, sampling rates, synchronization methods, filters applied, reproducibility of measurements?
What is the mean frequency ratio? How was it determined?
It is not clear what RSI measures and why is it essential in the context of this research. Please report the reproducibility of this test in young subjects.
The method used to measure leg stiffness has poor validity.
The statistical analysis used is completely unacceptable. Repeated measures ANOVA or similar should be used instead (read and apply PubMed ID: 31149752).
Minor
L40. References.
L41-43. Any exercise may cause fatigue. If this is important, you should be more precise (or delete).
L47. What neuromuscular functions (be more precise)
L61 “are an at risk”
L77 How was serious injury defined?
L84-85. How and when was the information necessary to calculate PHV collected?
L90. “The players were tested repeated-measures design”
Many more…